# Computational Model for Episodic Timeline Based on a Spectrum of Synaptic Decay Rates

**James Mochizuki-Freeman, Sara Zomorodi, Sahaj Singh Maini, Zoran Tiganj**

Department of Computer Science, Indiana University Bloomington

## Abstract

**Human episodic memory enables the retrieval of temporally organized past experiences. Retrieval cues can target both semantic content and temporal location, reflecting the multifaceted nature of episodic recall. Existing computational models provide mechanistic accounts for how temporally organized memories of the recent past (seconds to minutes) can persist in neural activity. However, it remains unclear how episodic memories can be stored and retrieved while preserving temporal structure within and across episodes. Here, we propose a computational model that uses a spectrum of synaptic decay rates to store temporally organized memories of the recent past as an episodic timeline. We characterize how the memories can be retrieved using either a memory of the recent past, specific semantic cues, or temporal addressing. This approach thus bridges short-term working memory and longer-term episodic storage, offering a computational model of how synaptic dynamics can maintain temporally structured events.**

**Keywords:** Episodic Memory; Working Memory; Laplace Transform; Associative Learning

## Introduction

The ability to encode and retrieve episodic memories is a critical faculty of human cognition. Episodic memories are commonly characterized by vividly remembering a particular experience, including the temporal organization of the events during the episode (Tulving, 2002, 1993; Dickerson & Eichenbaum, 2010; Hasselmo, 2013; Moscovitch et al., 2016; Ranganath, 2024; Michaelian, 2024). The retrieval process itself involves a combination of intertwined semantic and temporal cues. For instance, imagine being in the aisle of a supermarket chain trying to remember where to find the milk. Assuming you have been to multiple stores belonging to the same chain and the location of the milk varies from store to store, you might have to rely on either semantic or temporal cues. If you rely on semantic cues, you might look around and gather more information about the particular store to probe your memory and retrieve the required information. If you use episodic memory, you might try to isolate a particular experience of being in that store and use that memory to retrieve the location of the milk. To retrieve a particular experience, you might use semantic cues from your surroundings or temporal cues, such as recalling being in that store a month ago. All of these strategies might lead to a successful outcome. To further illustrate the importance of temporal retrieval of episodic memory, imagine waking up in a hotel room at night. Even in complete darkness, with no cues about your location or the day, you can recall recent events. This suggests that memories are temporally organized, enabling retrieval based on temporal recency, even beyond the scope of short-term neural activity.

The temporal organization of memory has been extensively studied in multiple experimental paradigms. The temporal contiguity effect in free recall (Kahana, 1996) shows that after recalling an item from the list, participants tend to recall an item that was in its temporal proximity during encoding. Computational models of this process commonly include binding of the gradually changing temporal context with the present input (Howard & Kahana, 2002; Polyn et al., 2009; B. Sederberg et al., 2010). In judgment of recency (Hacker, 1980; Hintzman, 2005), when comparing which of two probes was presented more recently, participants show response time profiles that depend only on the lag of a more recent probe (Hacker, 1980; Tiganj et al., 2022; Maini et al., 2022), consistent with the hypothesis that they are scanning a mental timeline of the past (Howard et al., 2015). Similarly, in a judgment of imminence task where, after learning a probabilistic sequence, participants choose which of two probes they believe will appear sooner, the response times depend on the lag on the more imminent probe. This is consistent with the hypothesis that the timeline of the past has been input to the associative learning mechanisms producing a timeline of the future (Tiganj et al., 2021). Critically, for both past and future, the response times grew sublinearly with the lag, resembling logarithmic compression, consistent with Weber-Fechner's law (Fechner, 1860; Portugal & Svaiter, 2011).

The role of time as a governing principle for organizing episodic memories has been extensively investigated in cognitive neuroscience studies. It has been reported that neural activity in the hippocampus during recall reinstates the neural activity observed during encoding, indicating the retrieval of temporal context through a neural "jump back in time" (Howard et al., 2012; Folkerts et al., 2018; Lohnas et al., 2023; Broitman & Kahana, 2024). Research on other mammals has also revealed the ability to jump back in time and retrieve temporal context, much like humans (Crystal & Suddendorf, 2019; Crystal, 2024).

Here we propose a computational model that unifies these different properties of human memory and extends them across a wide range of temporal scales, from seconds to days. Specifically, the proposed model describes (1) how temporally organized sequences of recent past can be stored in episodic memory (2) how such memory representation can be retrieved using either semantic or temporal cues.

## Related work and background: Constructing a log-compressed memory timeline of the recent past using Laplace and inverse Laplace transform

Our approach builds on prior modeling work, which proposed that memory of the recent past can be maintained through gradually changing neural activity (leaky integrators with a spectrum of time constants), giving rise to sequences of neural activity that encode a timeline of what happened when over the recent past (Shankar & Howard, 2012; Howard et al., 2014). The fundamental idea behind this approach is that given a set of neurons with exponentially decaying impulse responses with a spectrum of time constants, their instantaneous firing rates will encode an approximation of the real-domain Laplace transform. Thus, the input function over time is encoded into the instantaneous firing rates of a set of neurons.

By inverting this Laplace transform, we can recover a temporally structured representation of the input history. The Laplace transform can be inverted through a linear transformation, converting neurons with exponentially decaying firing rates into neurons with sequentially activated, bell-shaped impulse responses. These sequentially activated neurons constitute a time-addressable metric space in the sense that each neuron is characterized by the peak time of its impulse response, hence coding an approximation of the input function that much time in the past. Critically, the model gives rise to scale-invariant sequential activations: neural activations are rescaled versions of each other—each stretched proportionally to its peak time. Time constants of the exponential decay are chosen such that the resulting inverse transform has logarithmically spaced peaks. The scale-invariance of the impulse response and the log-spacing of the peaks implies that the memory representation itself is logarithmically compressed. When observed across the log axis, it is supported by equally wide and equidistant sequential activations. This means that the more recent past is encoded with higher temporal resolution than the more distant past (Fig. 1).

Previous modeling work also addressed building associative memory using the log-compressed timeline (Shankar & Howard, 2012). Specifically, the associative memory stored in synaptic weights binds the temporal context encoded within the log-compressed timeline with the current input. After learning, this associative memory representation can be probed with the current temporal context, resulting in the prediction of the next stimulus. Furthermore, the Laplace transform can be translated into the future by modulating the recurrent weights of exponentially decaying cells: this can speed up the decay enabling the system to compute the future values of the memory representation (assuming no new inputs) (Shankar et al., 2016; Tiganj, Gershman, et al., 2019; Goh et al., 2022). Importantly, since it mirrors the past, the prediction is also log-compressed. This means the more imminent future is represented with a higher temporal resolution than the more distant future. These properties enabled previous modeling work to account for effects observed in a variety of

behavioral tasks including serial scanning in temporal judgment tasks (Tiganj et al., 2021; Tiganj, Cruzado, & Howard, 2019; Howard et al., 2015) and recency and contiguity effects in free recall (Howard et al., 2015).

The utility of the Laplace code for constructing a representation of the future has led to computational models of multi-scale reinforcement learning, where the representation of future time can be integrated to compute the estimated value of a particular choice (Tiganj, Gershman, et al., 2019; Momennejad & Howard, 2018; Tano et al., 2020; Masset et al., 2025; Howard et al., 2024; Sousa et al., 2025; Maini & Tiganj, 2025).

## Constructing an episodic timeline using a Laplace transform implemented through gradually changing synaptic weights

We extend the previous work by addressing fundamental limitations: associative memory averages past experiences in such a way that it is not possible to retrieve individual episodic memories. We propose a computational model for storing temporally organized episodic memories that also relies on the use of Laplace transform. However, in addition to using the Laplace transform to store memory of the recent past in neural activity, we also use it to store the history of those memories using synaptic weights. We achieve this by having associative weights decay with a spectrum of time constants, encoding the history of changes in the associative memory. We illustrate how inverting the Laplace transform results in the log-compressed timeline of episodic memories where individual episodic memories can be accessed using temporal or semantic cues.

## Computational Model

In this section, we present a computational model that unifies working and episodic memory by leveraging multi-scale synaptic dynamics.[1] We begin by showing how a set of neurons with exponentially decaying impulse responses can implement a log-compressed representation of the recent past (the "working memory timeline") via a Laplace transform. Next, we describe how associative learning linking this working memory representation to current inputs, allows the system to capture and predict temporal relationships. By translating the memory representation in the working memory timeline into the future and iteratively probing the associative memory we create a timeline of the future. We extend this framework to store entire episodes by encoding changes in the associative weights themselves, effectively creating an "episodic timeline" such that individual episodes can be retrieved using either temporal or semantic cues.

### Working memory timeline

To build a neural representation of the recent past (i.e., working memory), we create a timeline: an ordered set of neurons where each neuron encodes the input value from a specific

---

[1]The code implementation can be found at: `https://github.com/cogneuroai/episodic-timeline`

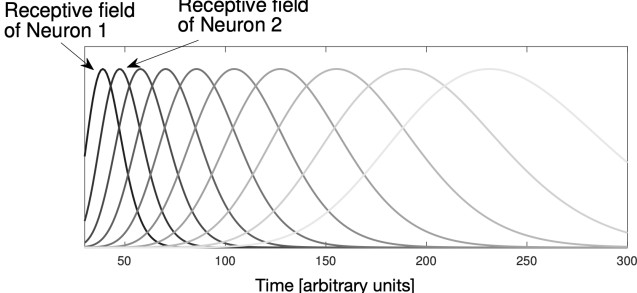
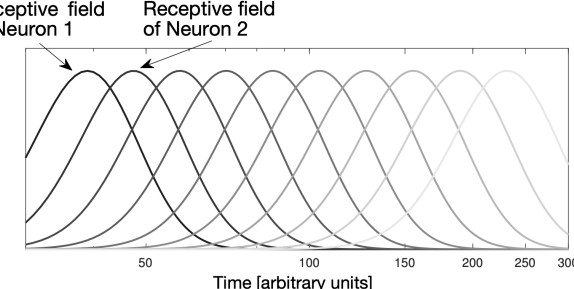

Figure 1: Log-compressed neural impulse responses. The responses are shown with respect to linear (left panel) and logarithmic (right panel) time axes. The peaks are log-spaced and the responses are scale-invariant: the width of the impulse responses increases linearly with their peak time. When displayed on a log-axis, the peaks are equidistant and the impulse responses are equally wide.

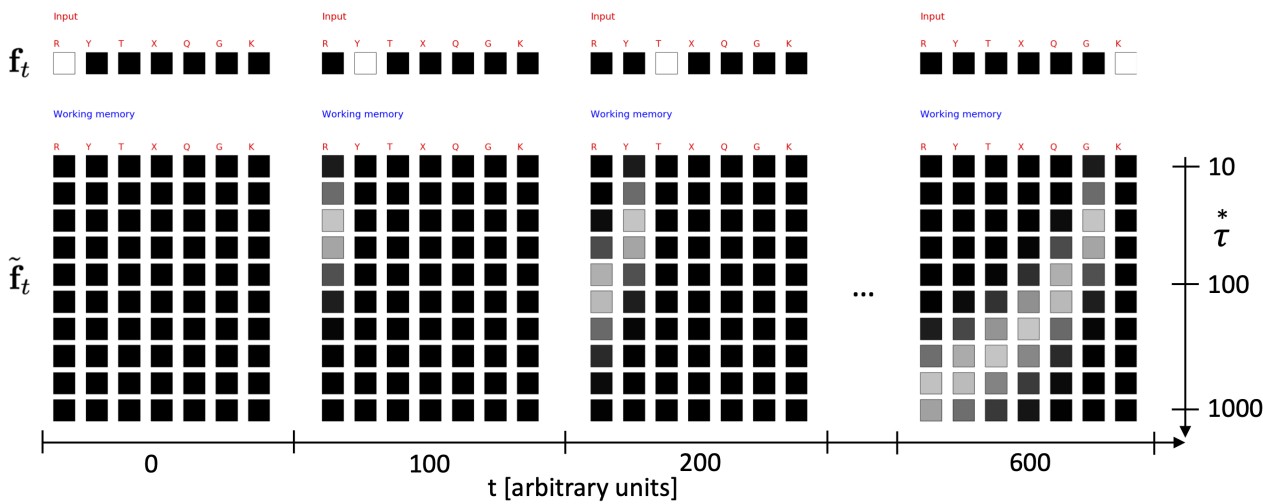

Figure 2: Illustration of the working memory timeline. An input sequence of 7 one-hot encoded letters is presented sequentially ($\mathbf{f}(t)$). The letter presentation is marked with the white color of the letter box. Each letter has a corresponding working memory representation consisting of sequentially activated neurons. Each box under a letter symbol represents an individual neuron in the matrix $\tilde{\mathbf{f}}(t)$. The level of activation is coded with shades of gray such that brighter shades represent higher activation. When the last letter is presented, $\tilde{\mathbf{f}}(t)$ contains a compressed timeline that stores what happened when. To appreciate the compression of the timeline, notice that the white trace of activity in the memory representation is curved and gradually spreads from letters that were presented more recently to letters that were presented less recently. While this example is made particularly simple to illustrate the concept, in general, $\mathbf{f}(t)$ does not have to be a one-hot vector, and its magnitude can take any value.

time in the past. More formally, given an input vector $\mathbf{f}(t)$ composed of $N$ elements ($i = 1...N$), the working memory is represented as a matrix $\tilde{\mathbf{f}}(t)$ of dimension $L \times N$, where $L$ is the number of units ($j = 1...L$) composing the timeline. At any time $t = t'$, $\tilde{\mathbf{f}}(t')$ holds a memory of the input signal, $\mathbf{f}(t < t')$. Rows $j$ of $\tilde{\mathbf{f}}(t)$ correspond to temporally smoothed values of $\mathbf{f}(t)$ in the past. While the past could be stored veridically such that the $j$-th row of $\tilde{\mathbf{f}}(t)$ stores the value of $\mathbf{f}(t)$ at a specific discrete step $t - j$ in the past, that would not be biologically plausible. Such memory would fundamentally differ from human memory, which is characterized by gradual decay and whose temporal resolution is governed by the Weber-Fechner law.

Hence, here we construct a scale-invariant logarithmically compressed memory of the recent past, using the same approach as Shankar & Howard (2012). This means that the $j$-th row of $\tilde{\mathbf{f}}(t)$ stores a temporally smoothed value of $\mathbf{f}(t)$ centered around $\overset{*}{\tau}_j$ time in the past. Scale-invariance of the representation is accomplished through the increasing width of the temporal windows (proportional to $\overset{*}{\tau}_j$) and having the centers of the windows $\overset{*}{\tau}_j$ be logarithmically spaced. This results in equal width and uniform spacing along the log-time axis (Fig. 1). Properties of such working memory in response to a temporal sequence of inputs are illustrated in Fig 2.

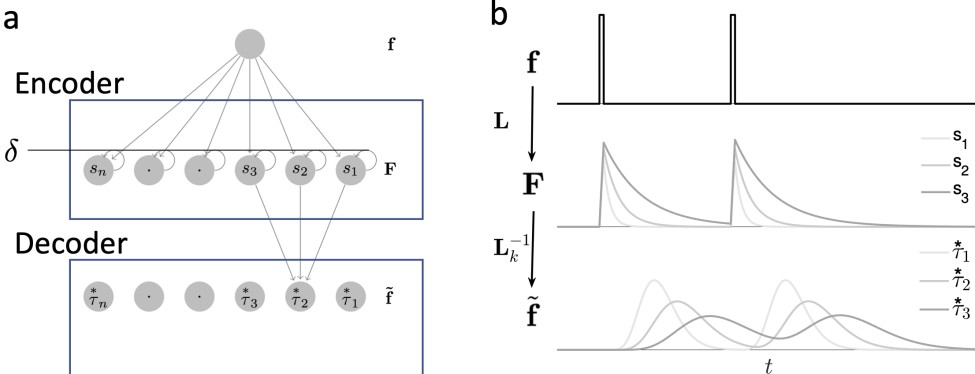

Figure 3: **(a)** Schematic of the working memory network. **(b)** Response of the neurons in the network to two brief pulses. Input signal $f_i(t)$ arrives into a set of recurrent nodes $F_{j,i}(t)$ whose states approximate the Laplace transform $F_i(s)$. The units operate on a spectrum of analytically computed rate constants $s_j$. A second layer inverts the transform and outputs a scale-invariant timeline $\tilde{f}_{j,i}(t)$. Neurons in the output layer are activated sequentially with the receptive fields increasing in a scale-invariant way. This means that at any moment $t'$, the activity of the neurons in the output layer encodes a compressed temporal history of the input signal (note that the input signal can be any function of time, brief pulses are chosen for simplicity). The output representation has a metric structure (each unit $j$ has its fixed place in the sequence, corresponding to time $\overset{*}{\tau}_j$ in the past).

To construct scale-invariant log-compressed $\tilde{\mathbf{f}}(t)$ in a time-local fashion (meaning without the need to explicitly store past values in a buffer), we use a recurrent neural network followed by a feedforward layer. Fig. 3a illustrates the structure of the network for a single input feature $f_i(t)$. The input signal $f_i(t)$ is fed into a set of $L$ recurrent units $F_{j,i}(t)$ that encode an approximation of the Laplace transform, which we denote collectively as $\mathbf{F}_i(t)$. The transform is implemented through units acting as leaky integrators with different decay rates $s_j$. In continuous time, the dynamics are:

$$\frac{dF_{j,i}(t)}{dt} = -s_j F_{j,i}(t) + f_i(t) \qquad (1)$$

This uses fixed, analytically computed parameters (the decay rates $s_j$), avoiding biologically non-plausible learning rules like backpropagation through time as in standard RNNs. For simulations using discrete time steps $\Delta t$, this differential equation is implemented as:

$$F_{j,i}(t) = (1 - s_j \Delta t)F_{j,i}(t - \Delta t) + f_i(t)\Delta t \qquad (2)$$

Here $j = 1...L$ indexes the Laplace units, $i = 1...N$ indexes the input features, and $\mathbf{s} \in \mathbb{R}^L$ is a vector composed of $L$ log-spaced decay rate values $s_j$. The dimension of the state matrix $\mathbf{F}(t)$ is $L \times N$, the same as $\tilde{\mathbf{f}}(t)$.

The Laplace transform can be inverted into $\tilde{\mathbf{f}}(t)$ through a feedforward network (denoted as operator $\mathcal{L}^{-1}$) with analytically computed weights. For instance, this can be done using Post's inversion formula, with a fixed integer $k$:

$$\tilde{f}_{j,i}(t) \equiv \mathcal{L}^{-1}[F_{j,i}(t)] = \frac{(-1)^k}{k!}\left(\frac{k}{\overset{*}{\tau}_j}\right)^{k+1} \frac{d^k F_{\cdot,i}(\cdot,s)}{ds^k}\bigg|_{s=k/\overset{*}{\tau}_j} \qquad (3)$$

where $F_{\cdot,i}(\cdot, s)$ represents the analytically continued Laplace transform derived from the states $F_{j,i}(t)$, and $\overset{*}{\tau}_j$ is related to the $j$-th decay rate ($s_j = k/\overset{*}{\tau}_j$) and defines the center of the temporal receptive field for the $j$-th timeline unit.

Critically, while $\tilde{\mathbf{f}}(t')$ and $\mathbf{F}(t')$ contain the same amount of information about $\mathbf{f}(t < t')$, $\tilde{\mathbf{f}}(t')$ represents this information in the form of a metric space (indexed by $j$, corresponding to time $\overset{*}{\tau}_j$ in the past), making navigation through time analogous to navigation through neural space spanned by $\tilde{\mathbf{f}}$. We note that alternatively to Post inversion formula, several other techniques to invert the Laplace transform exist, including CME (Horváth et al., 2020), CME-R (Mészáros & Telek, 2022), and the Gaver-Stehfest method (Gaver, 1966; Stehfest, 1970), often offering better numerical stability (see Appendix for further discussion on alternative approaches to computing the inverse).

The impulse response (response to input $f_i(t') = \delta(t)$) of the $j$-th timeline unit for input $i$, $\tilde{f}_{j,i}(t)$ is given by:

$$\tilde{f}_{j,i}(t) = \frac{k^{k+1}}{k!}\frac{1}{t}\left(\frac{t}{\overset{*}{\tau}_j}\right)^{k+1} e^{-k\frac{t}{\overset{*}{\tau}_j}} \quad \text{(for } t > 0) \qquad (4)$$

The peak time of $\tilde{f}_{j,i}(t)$ can be found by taking the partial derivative with respect to $t$, equating it to zero, and solving for $t$: $\partial \tilde{f}_{j,i}(t)/\partial t = 0 \rightarrow t = \overset{*}{\tau}_j$. Therefore, each neuron $j$ in $\tilde{\mathbf{f}}(t)$ peaks at $\overset{*}{\tau}_j$, making $\overset{*}{\tau}_j$ effectively an internal representation of time in the past. Fig. 4 shows an example of an input sequence composed of orthogonal stimuli (top row) encoded in $\tilde{\mathbf{f}}(0)$ (bottom row). The width of the temporal receptive fields decreases with $k$ — large $k$ makes the inverse more precise resulting in narrower impulse response, but leading to larger sensitivity to noise in $\mathbf{F}$ when computing the inverse (see Ap-

pendix for further discussion on the impact of noise).

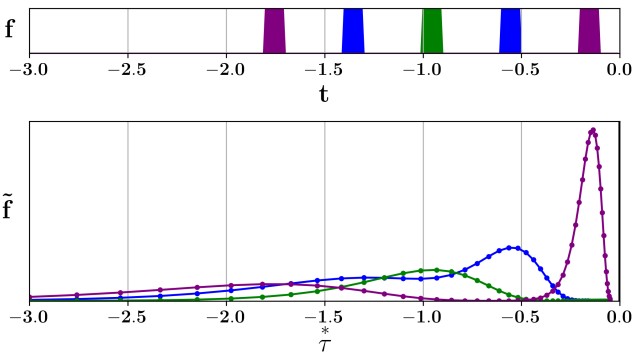

Figure 4: Illustration of working memory storing a sequence of orthogonal stimuli. At time $t = 0$, memory trace $\tilde{\mathbf{f}}(0)$ (bottom row) encodes a log-compressed history of the input sequence (top row). Note the higher resolution for the more recent past. Each dot in the bottom row corresponds to the activity of an individual neuron $\tilde{f}_{j,i}(0)$, plotted relative to its peak time $\overset{*}{\tau}_j$.

## Learning temporal relationships via associative memory

We use Hebbian associative memory to construct a three-dimensional tensor $\mathbf{M}(t)$ that stores associations between the input $\mathbf{f}(t)$ and the memory of the recent past $\tilde{\mathbf{f}}(t)$. We define $\mathbf{M}(t)$ as an $N \times N \times L$ tensor, where $M_{i',i,j}(t)$ represents the association from input feature $i$ at timeline position $j$ (time $\overset{*}{\tau}_j$ ago) to current input feature $i'$. Change in $\mathbf{M}(t)$ at every time step (in discrete time $\Delta t$) can thus be written as an outer product between $\mathbf{f}(t)$ and $\tilde{\mathbf{f}}(t)$:

$$\Delta M_{i',i,j}(t) = f_{i'}(t)\,\tilde{f}_{j,i}(t) \tag{5}$$

This change represents the association formed at time $t$. It is then accumulated into the associative memory tensor:

$$\mathbf{M}(t) = \mathbf{M}(t - \Delta t) + \Delta \mathbf{M}(t) \tag{6}$$

When probed with the memory representation of the recent past $\tilde{\mathbf{f}}(t)$ at time $t$, $\mathbf{M}(t)$ provides a prediction $\mathbf{p}(t)$ of the input vector:

$$p_{i'}(t) = \sum_{i=1}^{N} \sum_{j=1}^{L} M_{i',i,j}(t)\,\tilde{f}_{j,i}(t) \tag{7}$$

In order to predict more than one step into the future, we perform time translation of the working memory towards the future, effectively constructing an estimate of the future state of temporal context $\tilde{\mathbf{f}}(t + \delta)$. To achieve this, we use the fact that time translation is easily doable in the Laplace domain. It is based on the fact that the impulse response of $F_{j,i}(t)$ is an exponentially decaying function ($e^{-s_j t}$). If the input $f_i(t)$ becomes zero after time $t$, we can compute the time-translated value of $F_{j,i}(t)$ at time $t + \delta$ by applying the decay:

$$F_{j,i}(t + \delta) = e^{-s_j \delta} F_{j,i}(t) \tag{8}$$

This computation can be performed using a diagonal matrix $\mathbf{R}^\delta$ where $R_{j,j}^\delta = e^{-s_j \delta}$, such that the vector $\mathbf{F}_i(t + \delta) = \mathbf{R}^\delta \mathbf{F}_i(t)$.

Consequently, we can compute a time-translated version of the working memory $\tilde{\mathbf{f}}(t + \delta)$ by applying the inverse Laplace transform operator (Eq. 3) to the time-translated Laplace state $\mathbf{F}(t + \delta)$.

If we now probe the current associative memory $\mathbf{M}(t)$ with the memory representation translated into the future $\tilde{\mathbf{f}}(t + \delta)$, we obtain a prediction $\mathbf{p}(t + \delta)$ of the input at $\delta$ time in the future:

$$p_{i'}(t + \delta) = \sum_{i=1}^{N} \sum_{j=1}^{L} M_{i',i,j}(t)\,\tilde{f}_{j,i}(t + \delta) \tag{9}$$

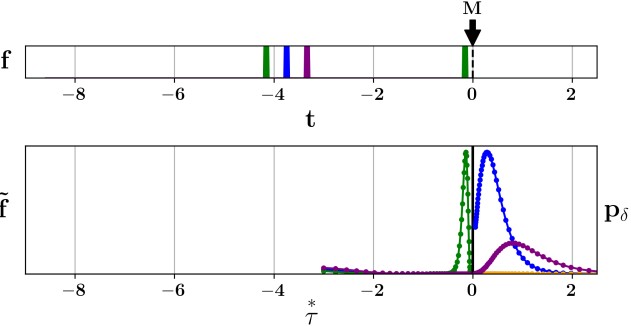

Figure 5: Illustration of prediction based on associative memory $\mathbf{M}$. Since in this example the input $f$ consists of a single sequence, associative memory $\mathbf{M}$ is sufficient for successful prediction. The associative memory is probed following the repetition of the green stimulus (location marked with the dashed line and letter 'M' in the top row). The negative side of the bottom plot corresponds to the memory trace $\tilde{\mathbf{f}}$, while the positive side corresponds to the prediction $\mathbf{p}_\delta$. While the green stimulus is in the memory, prediction unfolds first for blue (which has a relatively high temporal resolution) and then for the temporally more distant purple stimulus.

An example of constructing $\mathbf{p}(t + \delta)$ is provided in Fig. 5. A three-dimensional one-hot input is presented in a sequence (top row). The sequence is used to compute $\mathbf{M}(t)$ as described in Eq. 6. After the first element of the sequence is presented again, $\mathbf{p}(t + \delta)$ is computed following Eq. 9 for a set of log-spaced $\delta$ values (bottom row). $\mathbf{p}(t + \delta)$ successfully predicts the time of occurrence of the following two stimuli with a temporal resolution that gradually decays (due to log-spaced $\delta$). The amplitude of the peaks of the prediction falls as power-law function of $\overset{*}{\tau}_j$: $1/\overset{*}{\tau}_j$.

To illustrate the limitations of predicting using $\mathbf{M}(t)$ that occur due to averaging, we created an example with two sequences, each composed of three orthogonal inputs (Fig A1). The first element of each of the two sequences was the same. That element was later repeated (third time), and following the repetition, we computed $\mathbf{p}(t + \delta)$. The resulting prediction

now predicts elements from both sequences — the prediction traces overlap. This illustrates that only associative memory $\mathbf{M}(t)$ is not sufficient to recover individual episodes.

## Episodic memory timeline

We create an episodic memory timeline by encoding the history of changes in associative memory weights. This timeline captures when each memory association was modified, enabling the retrieval of specific episodes. To achieve this, we use the same Laplace framework, but this time with the changes in the associative strengths $\Delta\mathbf{M}(t)$ (Eq. 5) as inputs to the leaky integrators which compute a log-spaced approximation of the Laplace transform, $\mathbf{E}(t)$. Here $\mathbf{E}(t)$ integrates $\Delta\mathbf{M}(t)$ using leaky integrators with very slow decay rates $\sigma_l$. In continuous time, the dynamics for each element $(i', i, j)$ and each episodic decay rate index $l$ (where $l = 1...L_e$) would be:

$$\frac{dE_{i',i,j,l}(t)}{dt} = -\sigma_l E_{i',i,j,l}(t) + \Delta M_{i',i,j}(t) \qquad (10)$$

where $\sigma \in \mathbb{R}^{L_e}$ is a vector composed of $L_e$ log-spaced values $\sigma_l$ that determine the rate of decay of units in $\mathbf{E}(t)$. Since $\Delta M_{i',i,j}(t)$ is computed in discrete steps based on $\tilde{\mathbf{f}}(t)$, the practical implementation uses discrete time steps $\Delta t$:

$$E_{i',i,j,l}(t) = (1 - \sigma_l \Delta t) E_{i',i,j,l}(t - \Delta t) + \Delta M_{i',i,j}(t) \qquad (11)$$

Here, $\mathbf{E}(t)$ is a four-dimensional tensor, with dimensions $N \times N \times L \times L_e$. Unlike working memory, which stores memory representations over a relatively short time (spanning seconds to minutes), $\mathbf{E}(t)$ stores changes in the weights with large time constants $\tau'_l \approx 1/\sigma_l$, spanning days and beyond (see discussion on biological plausibility of slow synaptic changes). Critically, because $\sigma$ contains logarithmically spaced rates, this representation can be constructed with resources that scale with the logarithm of the largest temporal scale.

We can then compute the inverse Laplace transform $\tilde{\mathbf{M}}(t)$ similarly as for the working memory. Applying Post's inversion formula (using fixed $k$) along the last dimension ($l = 1...L_e$) yields the episodic timeline tensor:

$$\tilde{M}_{i',i,j,l}(t) \equiv \mathcal{L}^{-1}[E_{i',i,j,l}(t)]$$
$$= \frac{(-1)^k}{k!} \left(\frac{k}{\tau'_l}\right)^{k+1} \frac{d^k E_{i',i,j,\cdot}(\cdot,\sigma)}{d\sigma^k}\Bigg|_{\sigma=k/\tau'_l} \qquad (12)$$

Here $\tau'_l = k/\sigma_l$ represents the characteristic center time of the temporal receptive field for the $l$-th unit of the episodic timeline. $\tilde{\mathbf{M}}(t)$ is an $N \times N \times L \times L_e$ tensor, where $\tilde{M}_{i',i,j,l}(t)$ represents the associative strength $(i, j) \rightarrow i'$ formed around $t - \tau'_l$ time in the past.

Having episodic memory $\tilde{\mathbf{M}}(t)$, it is possible to use a content- or time-addressable approach to "jump back in time" and retrieve the episodic memory corresponding to a specific past time. This involves selecting a specific index $\varepsilon \in \{1...L_e\}$ that corresponds to the desired time point $\tau'_\varepsilon$ in the past. Given the retrieved memory slice $\tilde{\mathbf{M}}^{(\varepsilon)}(t)$ (where the last index $l$ is

fixed to $\varepsilon$), the predictions $\mathbf{p}^\varepsilon(t + \delta)$ can be constructed in a similar way as before (Eq. 9), but using the retrieved episodic associations:

$$p_{i'}^\varepsilon(t + \delta) = \sum_{i=1}^{N} \sum_{j=1}^{L} \tilde{M}_{i',i,j,\varepsilon}(t) \, \tilde{f}_{j,i}(t + \delta) \qquad (13)$$

where index $\varepsilon$ denotes the specific temporal location along the episodic timeline being retrieved.

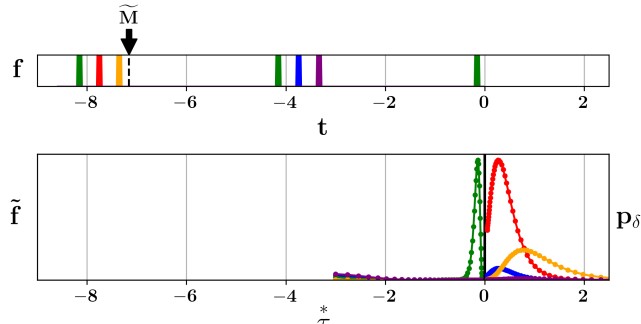

Figure 6: Illustration of prediction based on episodic memory $\tilde{\mathbf{M}}$. The input consists of two different sequences (first consists of green followed by red followed by yellow and the second consists of green followed by blue followed by purple) and a repetition of the green stimulus. The prediction is made at time 0, following the repetition of the green stimulus. The arrow in the top row indicates the location of "jump back in time" (selected index $\varepsilon$) where the episodic memory is selected. This results in temporal separation of the two sequences and prediction of the elements from the first sequence (red followed by yellow). Interference is rather small since the amplitudes of the blue and purple traces (which belong to the second sequence, far from the selected index $\varepsilon$) are much smaller than amplitudes of the red and yellow traces (which are close to the selected index $\varepsilon$).

We demonstrated retrieval of episodic memories and resulting predictions $\mathbf{p}^\varepsilon(t + \delta)$ on several examples, similar to the examples we investigated earlier. In Fig 6, we show two sequences of three orthogonal stimuli, with the first stimulus being common for each of the two sequences and then repeated for the third time. The prediction is generated when episodic memory is retrieved from the end of the first sequence (i.e., selecting $\varepsilon$ corresponding to $\tau'_\varepsilon$ around that time): following the stimulus repetition, it retrieves the remaining items in the first sequence.

Similarly, when episodic memory is retrieved soon after the second sequence (selecting the appropriate $\varepsilon$), the prediction produces elements from the second sequence, demonstrating good temporal isolation of different episodes (Fig A2). We further tested this with sequences composed of 7 stimuli (Fig 7). While the high proximity of the two sequences and a relatively large number of stimuli is causing some interference, the elements from the first sequence (the center of episodic retrieval,

selected ε, in this example) have a much stronger presence in the retrieved sequence than the elements from the second sequence (note that $1/\overset{*}{\tau}_j$ attenuation of the amplitude in $\tilde{\mathbf{f}}(t+\delta)$ makes it more difficult to distinguish relative amplitudes of the elements later in the sequence).

To illustrate the properties of the model for non-orthogonal, more realistic examples, we conducted an experiment where the inputs to the model were CLIP (Radford et al., 2021) embeddings of video frames (we used LAION-5B (Schuhmann et al., 2022) pretrained CLIP embeddings). Specifically, to emphasize the capability of separating individual episodes, we designed the video to include two episodes with overlapping contexts (Fig. 8 and Fig. A3). The video consisted of 990 frames, each embedded as a 512-element long vector. Each episode was 215 frames long. Vector $\tau'_l$ consisted of 20 log-spaced values from 5.0 to 80 seconds (corresponding to 60 to 955 frames), and vector $\overset{*}{\tau}$ consisted of 16 log-spaced values from 1.0 to 12 seconds (corresponding to 12 to 143 frames). Probing $\tilde{\mathbf{M}}$ with that context and the particular time stamp (e.g., trying to remember where did you go for vacation a year ago) results in correct retrieval of the target location. This was not the case when probing the associative memory $\mathbf{M}$ since it averages individual episodes and cannot utilize the temporal information.

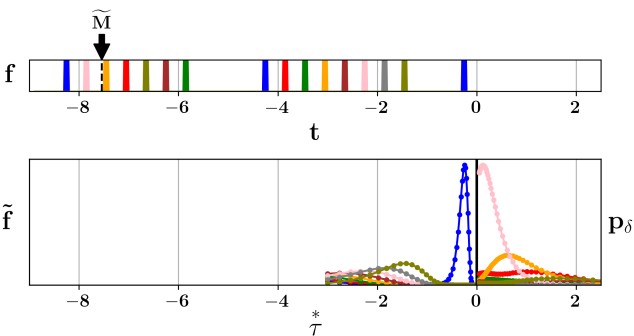

Figure 7: Illustration of prediction based on episodic memory $\widetilde{\mathbf{M}}$ for a long sequence of seven orthogonal stimuli. After blue stimulus is repeated for the third time (a little before time 0), at time 0, the episodic memory is retrieved from within the first sequence (ε corresponds to time within first sequence). The prediction mostly reflects the content of the first sequence. Some interference from the second sequence is also noticeable due to temporal proximity of the second sequence to ε.

## Biological Plausibility and Implementation

A key assumption of our model is the availability of synaptic processes that operate over a wide, logarithmically-spaced spectrum of timescales. This aligns with biological observations. For instance, synaptic plasticity mechanisms like long-term potentiation (LTP) are known to unfold across multiple phases, from early forms lasting minutes to hours, to later,

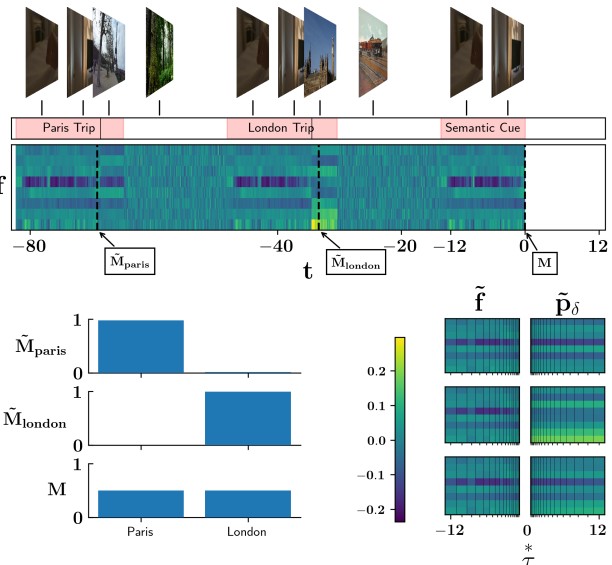

Figure 8: Illustration of prediction based on associative memory $\mathbf{M}$ and episodic memory $\widetilde{\mathbf{M}}$ for a sequence of non-orthogonal stimuli derived from CLIP embeddings of egocentric video frames. The input to the model consists of two distinct episodes with partially overlapping contexts. The top row shows several representative frames with indicators for two episodes (Paris and London trip). The second row shows a part (8 out of 512 elements) of the CLIP vectors used as input $\mathbf{f}$ to the model. In the first episode, the video input starts with a dark hotel room with a closed curtain, and the person gets up from the bed and opens the curtain to reveal the cityscape of Paris. In the second episode, everything is the same except that the revealed cityscape is the one of London. The two episodes are spaced by random images. After more random frames, the temporal context of the dark hotel room is repeated (semantic cue) and it becomes part of the working memory timeline ($\tilde{\mathbf{f}}$) that is used to probe the episodic timeline ($\tilde{\mathbf{M}}$). The three rows below the input demonstrate that by providing temporal context and a temporal pointer, the episodic timeline is able to identify a correct episode. First, probing $M$ with the temporal context reveals a mixed prediction of both Paris and London since $\mathbf{M}$ simply averages distinct episodes. However, probing $\tilde{\mathbf{M}}$ with the temporal context and specifying a temporal pointer to a time point near Paris or London trip (e.g., "Where was that trip I took a year ago" and selecting appropriate index ε corresponding to one year in the past), results in the prediction of the correct city. The plots show the temporal context of the recent past $\tilde{\mathbf{f}}$ and prediction $\mathbf{p}_\delta$ as a function of the internal log-compressed timeline spanned by $\overset{*}{\tau}$ as well as a bar plot indicating the average magnitude of the prediction for each of the cities. See Fig. A3 for a visualization of the prediction built using each $\tau'_l$ of the episodic timeline.

protein-synthesis-dependent forms persisting for days or even weeks (Frey et al., 1988; Abraham, 2003). Evidence sug-

gests this may reflect a continuum of plasticity decay rates rather than just two discrete categories, potentially supported by dynamic processes like receptor trafficking (Dong et al., 2015). This spectrum of timescales provides biological motivation for the spectrum of decay rates $\sigma_l$ underpinning the episodic timeline ($\tilde{\mathbf{M}}$).

Addressing how specific past episodes are 'read out' from the slow synaptic traces could correspond biologically to the reactivation of a specific memory engram or cell assembly formed during that past episode (Rao-Ruiz et al., 2021; Josselyn & Tonegawa, 2020). Such engrams, potentially involving functionally connected cell ensembles across brain regions, are thought to undergo persistent changes during encoding and be reactivated by relevant cues during retrieval. The subsequent probing step (Eq. 13) can then be interpreted as the influence of this reactivated engram's specific synaptic weight configuration ($\tilde{\mathbf{M}}$) on current neural activity patterns ($\tilde{\mathbf{f}}$), leading to pattern completion or prediction based on that unique past context.

In addition to biological support for diverse associative timescales, the core computational mechanisms underlying the Laplace transforms also align with plausible biological implementations. The required leaky integrators with long, exponentially decaying time constants ($\mathbf{F}$ nodes) are consistent with neurons exhibiting decaying persistent firing, potentially supported by specific ionic currents like the CAN current (Tiganj et al., 2015). Additionally, detailed circuit modeling has shown how the inverse Laplace transform operation ($L_k^{-1}$), required to produce the timeline representations ($\tilde{\mathbf{f}}$), can be instantiated in a network using specific patterns of local excitatory and inhibitory connections that respect Dale's Law (Liu et al., 2019). Recent work suggests that these transform-approximating weights could potentially emerge through self-supervised learning, particularly under constraints like local connectivity and shared synaptic weights (Alipour et al., 2025).

## Discussion

We developed a computational model of memory that unifies working memory and episodic memory into a common mathematical framework. The central innovation is representing both working and episodic memories as timelines operating on different timescales. Working memory encodes a log-compressed representation of recent inputs by way of sequentially activated neural activity that approximates the Laplace and the inverse Laplace transforms of the input sequence. Critically, these short-lived traces allow the system to learn, via Hebbian synaptic changes, temporal associations—what happened *when*—over seconds to minutes. Episodic memory is then formed by tracking the history of the changes in those synaptic weights, creating a longer-lasting record of which associations were formed and *when* they were formed.

By storing the history of weight changes rather than just the final weights, the model can retrieve an *individual* episode rather than a superposition or average of multiple experiences. Traditional associative memories often collapse overlapping inputs into a fused representation, making it difficult to isolate the specific sequence of events from a single occurrence. Here, the episodic timeline decouples one episode from another, because each set of weight changes is stamped with its own time signature via a spectrum of slow decays. The result is an ability to recover which episode occurred and in what chronological order.

The nature of memory storage and capacity in this Laplace-based timeline model differs fundamentally from attractor network models, such as classic or modern Hopfield networks (Hopfield, 1982; Krotov & Hopfield, 2016; Ramsauer et al., 2020). Attractor networks define capacity primarily by the number of distinct patterns that can be stored as stable states and retrieved via convergence based on content cues. Interference typically arises from pattern overlap, and exceeding capacity can lead to catastrophic forgetting or spurious states (Grossberg, 2020; Amit & Amit, 1989). In contrast, the proposed timeline model does not store discrete patterns as attractors. Its 'capacity' is reflected in the temporal resolution of the stored history; memory fidelity decreases systematically for events further in the past due to logarithmic compression, rather than abruptly failing when a specific number of patterns is exceeded. Interference is thus primarily temporal rather than based on pattern similarity alone, and retrieval relies on accessing specific points along the compressed temporal representation.

The model presented here builds directly upon frameworks that have successfully captured data in a variety of behavioral tasks, including temporal judgment tasks and free recall (Howard & Kahana, 2002; Howard et al., 2015; Tiganj et al., 2021). Specifically, the Hebbian associative memory component of our model (Eq. 6) is analogous to the associative structures used in prior temporal context and Laplace transform models. Our current work inherits capabilities of these models for experiments where the memory is tested shortly after list presentation. The key innovation here—the episodic timeline generates specific predictions and explains ability to recall individual lists when tasks are conducted over longer timescales (e.g., across hours or days). Formal simulations comparing the model to data from such extended recall paradigms remain an important direction for future work.

Despite its conceptual strengths, our model leaves several open questions. One limitation is that we have not fully explored capacity constraints or noise effects in large-scale implementations. Biological systems may use approximate versions of these transforms, leading to errors or interference in retrieving older memories. Additionally, we have not explicitly modeled the role of hippocampal-cortical interactions, which are known to underlie long-term consolidation processes. Integrating these processes—especially replay or offline consolidation—could provide a richer account of how short-term working memory traces become stabilized into lasting episodic memories.

## Acknowledgment

We gratefully acknowledge support from the Defense Advanced Research Projects Agency (DARPA) under project Time-Aware Machine Intelligence (TAMI) and the National Institutes of Health's National Institute on Aging, grant 5R01AG076198-02. This content is solely the responsibility of the authors and does not necessarily represent the official views of DARPA or the National Institutes of Health's National Institute on Aging. This research was supported in part by Lilly Endowment, Inc., through its support for the Indiana University Pervasive Technology Institute.

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

# Appendix

The appendix provides pseudocode detailing the core computations of the model for encoding and retrieval, supplemental figures, a discussion on robustness to noise and a table listing descriptions of all the notations used throughout the paper.

---

**Algorithm 1** Episodic Timeline Model: Initialization and time step update during encoding

---

**Input:** Input dimensions $N$, WM timeline units $L$, EM timeline units $L_e$, Post's inversion parameter $k$, WM decay rates $\mathbf{s} \in \mathbb{R}^L$, EM decay rates $\sigma \in \mathbb{R}^{L_e}$, time step $\Delta t$.

**Initialize:**

1: Set WM Laplace states $\mathbf{F}(0)$ to zero matrix ($L \times N$)
2: Set Associative memory $\mathbf{M}(0)$ to zero tensor ($N \times N \times L$)
3: Set EM Laplace states $\mathbf{E}(0)$ to zero tensor ($N \times N \times L \times L_e$)
4: Precompute Inverse Laplace operator weights $\mathbf{W}_{inv}$ based on $k$ and $\mathbf{s}$ for $\tilde{\mathbf{f}}$ (implementing Eq. 3)
5: Precompute Inverse Laplace operator weights $\mathbf{W}_{e,inv}$ based on $k$ and $\sigma$ for $\tilde{\mathbf{M}}$ (implementing Eq. 12)

6: **procedure** TIMESTEPUPDATE$(t, \mathbf{f}(t))$
  *// Update Working Memory Components*
7:   **for all** input features $i \in \{1..N\}$ **do**
8:     **for all** WM Laplace units $j \in \{1..L\}$ **do**
9:       $F_{j,i}(t) \leftarrow (1 - s_j \Delta t) F_{j,i}(t - \Delta t) + f_i(t) \Delta t$  ▷ Eq. 2
10:    **end for**
11:    Compute $\tilde{\mathbf{f}}_i(t)$ from $\mathbf{F}_i(t)$ using $\mathbf{W}_{inv}$ ▷ Apply Eq. 3
12:  **end for**
  *// Update Associative Memory*
13:  **for all** target features $i' \in \{1..N\}$ **do**
14:    **for all** source features $i \in \{1..N\}$ **do**
15:      **for all** WM timeline units $j \in \{1..L\}$ **do**
16:        $\Delta M_{i',i,j}(t) \leftarrow f_{i'}(t)\tilde{f}_{j,i}(t)$  ▷ Eq. 5
17:      **end for**
18:    **end for**
19:  **end for**
20:  $\mathbf{M}(t) \leftarrow \mathbf{M}(t - \Delta t) + \Delta \mathbf{M}(t)$  ▷ Eq. 6
  *// Update Episodic Memory Components*
21:  **for all** indices $(i', i, j)$ **do**  ▷ Dimensions $N \times N \times L$
22:    **for all** EM Laplace units $l \in \{1..L_e\}$ **do**
23:      $E_{i',i,j,l}(t) \leftarrow (1 - \sigma_l \Delta t) E_{i',i,j,l}(t - \Delta t) + \Delta M_{i',i,j}(t)$  ▷ Eq. 11
24:    **end for**
25:  **end for**
26: **end procedure**

---

**Algorithm 2** Model retrieval and prediction functions

---

1: **procedure** PREDICTFUTURE$(t, \delta)$  ▷ Predict input at time $t + \delta$
  *// Compute time-translated working memory state*
2:   Initialize $\tilde{\mathbf{f}}(t + \delta)$ (matrix $L \times N$)
3:   **for all** input features $i \in \{1..N\}$ **do**
4:     Initialize $\mathbf{F}_i(t + \delta)$ (vector $L \times 1$)
5:     **for all** WM Laplace units $j \in \{1..L\}$ **do**
6:       $F_{j,i}(t + \delta) \leftarrow e^{-s_j \delta} F_{j,i}(t)$  ▷ Eq. 8
7:     **end for**
8:     Compute $\tilde{\mathbf{f}}_i(t + \delta)$ from $\mathbf{F}_i(t + \delta)$ using $\mathbf{W}_{inv}$  ▷ Apply Eq. 3
9:   **end for**
  *// Probe associative memory $\mathbf{M}(t)$ with future state*
10:  Initialize prediction $\mathbf{p}(t + \delta)$ (vector $N \times 1$)
11:  **for all** target features $i' \in \{1..N\}$ **do**
12:    $p_{i'}(t + \delta) \leftarrow \sum_{i=1}^{N} \sum_{j=1}^{L} M_{i',i,j}(t)\tilde{f}_{j,i}(t + \delta)$  ▷ Eq. 9
13:  **end for**
14:  **return** $\mathbf{p}(t + \delta)$
15: **end procedure**

16: **procedure** PREDICTFROMEPISODE$(t, \varepsilon, \delta)$  ▷ Predict input at $t + \delta$ from episode $\varepsilon$
  *// Compute episodic memory state for selected episode $\varepsilon$*
17:  Initialize $\tilde{\mathbf{M}}^{(\varepsilon)}(t)$ (tensor $N \times N \times L$)  ▷ Represents $\tilde{M}_{i',i,j,\varepsilon}(t)$
18:  **for all** indices $(i', i, j)$ **do**  ▷ Dimensions $N \times N \times L$
19:    Compute $\tilde{M}_{i',i,j,\varepsilon}(t)$ from $E_{i',i,j,\cdot}(t)$ using $\mathbf{W}_{e,inv}$  ▷ Apply Eq. 12 for $l = \varepsilon$
20:    Assign $\tilde{M}_{i',i,j}^{(\varepsilon)}(t) \leftarrow \tilde{M}_{i',i,j,\varepsilon}(t)$
21:  **end for**
  *// Compute time-translated working memory state $\tilde{\mathbf{f}}(t + \delta)$*
22:  (Same computation as lines 3-9 in PREDICTFUTURE procedure)
23:  $\tilde{\mathbf{f}}(t + \delta) \leftarrow$ computed future WM state
  *// Probe retrieved episodic memory $\tilde{\mathbf{M}}^{(\varepsilon)}(t)$ with future state*
24:  Initialize prediction $\mathbf{p}^{\varepsilon}(t + \delta)$ (vector $N \times 1$)
25:  **for all** target features $i' \in \{1..N\}$ **do**
26:    $p_{i'}^{\varepsilon}(t + \delta) \leftarrow \sum_{i=1}^{N} \sum_{j=1}^{L} \tilde{M}_{i',i,j}^{(\varepsilon)}(t)\tilde{f}_{j,i}(t + \delta)$  ▷ Eq. 13
27:  **end for**
28:  **return** $\mathbf{p}^{\varepsilon}(t + \delta)$
29: **end procedure**

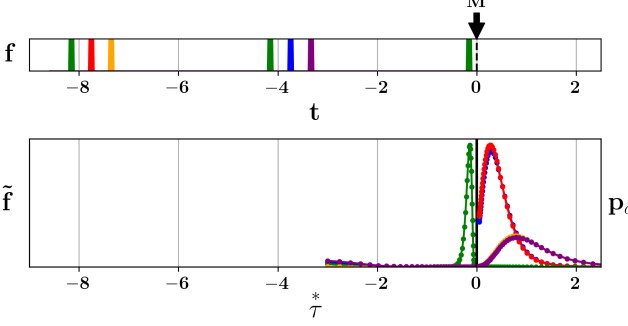

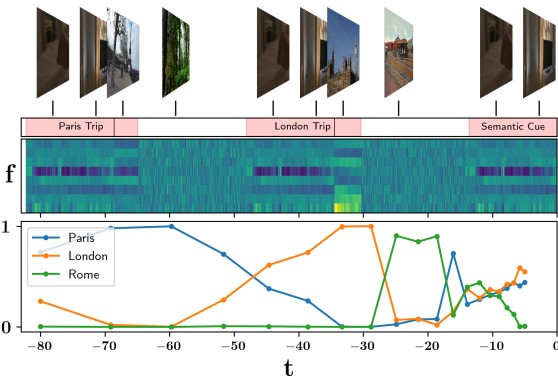

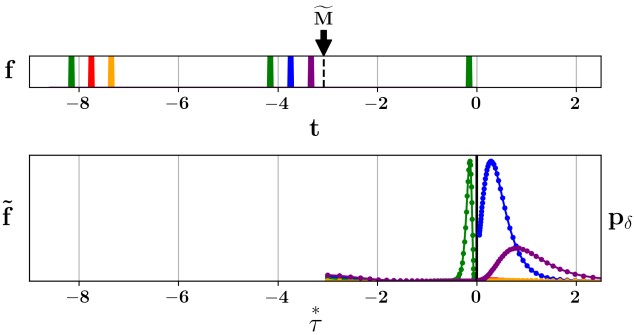

Figure A1: Illustration of prediction based on associative memory **M** for orthogonal stimuli. The input consists of two different sequences (first consists of green followed by red followed by yellow and the second consists of green followed by blue followed by purple) and a repetition of the green element. Associative memory **M** leads to the prediction of both sequences simultaneously (note the overlapping lines in the second row after time 0) without the ability to retrieve an individual sequence (individual episodic memory).

Figure A2: Illustration of prediction based on episodic memory $\widetilde{\mathbf{M}}$. Similar to Fig. 6, the arrow indicates the location of "jump back in time" ($\varepsilon$). The episodic memory contains ordered representation of the recent past (the content of working memory from that moment in the past) resulting in temporal isolation of the second sequence. Thus prediction is the strongest for the elements from the second sequence.

## Robustness, Noise Considerations, and Alternative Methods

The practical implementation and biological plausibility of the model depend on its robustness to noise and parameter variations. The inversion of the Laplace transform, particularly via Post's formula (Eq. 3, 12), can be sensitive to noise and requires careful implementation, including the choice of the parameter $k$ and the discretization of the decay rates ($s_j$ or $\sigma_l$).

The effects of noise in a similar linear system based on Laplace transforms were analyzed by Shankar & Howard

Figure A3: Illustration of prediction based on episodic memory for a sequence of non-orthogonal stimuli derived from CLIP embeddings of egocentric video frames. The input to the model consists of two distinct episodes with partially overlapping contexts. The top row shows several representative frames with indicators for two episodes (Paris and London trip). The second row shows part (8 out of 512 elements) of the CLIP vectors used as input **f** to the model. In the first episode, the video input starts with a dark hotel room with a closed curtain, and the person gets up from the bed and opens the curtain to reveal the cityscape of Paris. In the second episode, everything is the same except that the revealed cityscape is the one of London. The two episodes are spaced by random images. After more random frames, the temporal context of the dark hotel room is repeated (semantic cue) and becomes part of the working memory timeline ($\tilde{\mathbf{f}}$) that is used to probe the episodic timeline ($\tilde{\mathbf{M}}$). The last row indicates a prediction built in this way for each value of the log-spaced $\tau'_l$ (note the higher resolution for the more recent past). We display only the outputs of the prediction for the three cities, two of which were experienced in the previous episodes (Paris and London) and one that was not present in the input and served as a distractor (Rome). The predictions are normalized, so they sum up to 1. The prediction is near 1 for both Paris and London when $\tau'_l$ corresponds to the times around Paris and London trips.

(2012). Specifically, Shankar & Howard (2012) analyzed noise in a working memory model that only compute **M**, not the $\tilde{\mathbf{M}}$. However, since $\tilde{\mathbf{M}}$ is also computed using the Laplace and inverse Laplace transform, general properties regarding the impact of the noise still apply. The analysis from Shankar & Howard (2012) suggests several important properties relevant to this model:

**Input Noise:** Noise originating in the input layer ($\mathbf{f}(t)$), if correlated across the Laplace units (e.g., sensory noise affecting all $F_{j,i}$ for a given $i$ similarly), is transmitted linearly to the timeline representation ($\tilde{\mathbf{f}}(t)$ or $\tilde{\mathbf{M}}(t)$) without significant amplification.

**Internal Noise:** Uncorrelated noise affecting individual leaky integrator states ($F_{j,i}$ or $E_{i',i,j,l}$) can be amplified dur-

ing the inverse transform step (Eq. 3, 12), particularly if the discretization step ($\Delta s$ or $\Delta \sigma$) is small relative to the scale of noise variations. This highlights potential sensitivity to errors or fluctuations in the decay rates ($\sigma_l$ or $s_j$) or their spacing.

**Prediction Robustness:** Despite potential amplification in the timeline representation, Shankar & Howard (2012) showed that uncorrelated internal noise is strongly suppressed when computing the final prediction ($\mathbf{p}(t + \delta)$ or $\mathbf{p}^\varepsilon(t + \delta)$) via the associative memory readout (Eq. 9, 13). This suppression increases with the order $k$ of the Post's formula approximation, suggesting that the behavioral output (prediction) can be more robust than the internal timeline representation itself.

**Inversion Operator Noise:** Noise or imprecision in the weights implementing the inverse Laplace transform can directly affect the timeline representation and subsequent predictions. Stable noise might preserve timing but break scale invariance, while fluctuating noise could introduce variability in prediction timing. Methods such as Concentrated Matrix Exponential (CME) (Horváth et al., 2020), CME-R (Mészáros & Telek, 2022), and the Gaver-Stehfest method (Gaver, 1966; Stehfest, 1970) exist and may offer different trade-offs between accuracy, computational cost, and noise robustness in practical implementations.

Taken together, these findings suggest that while the internal representations might be sensitive to certain types of noise or parameter imprecision (like time-constant spacing), the associative readout mechanism provides a degree of robustness for predictions relevant to behavior.

| Symbol | Meaning / Dimensions |
|---|---|
| $N$ | Number of input features (length of $\mathbf{f}(t)$) |
| $L$ | Working-memory (WM) timeline units ($j = 1 \ldots L$) |
| $L_e$ | Episodic-memory (EM) timeline units – size of the *4th* dimension of $\tilde{\mathbf{M}}$ |
| $\mathbf{f}(t),\ f_i(t)$ | Input vector and its $i$-th component at time $t$ |
| $\tilde{\mathbf{f}}(t),\ \tilde{f}_{j,i}(t)$ | WM timeline matrix ($L{\times}N$) and its entry $(j,i)$ |
| $\mathbf{F}(t),\ F_{j,i}(t)$ | Laplace-layer states (leaky integrators), matrix $L{\times}N$ |
| $s_j$ | Decay rate of WM integrator $j$ |
| $\overset{*}{\tau}_j$ | Peak time of WM unit $j$; $\overset{*}{\tau}_j = k/s_j$ |
| $k$ | Order used in Post's inverse Laplace formula |
| $\mathbf{M}(t),\ M_{i',i,j}(t)$ | Associative-memory tensor ($N{\times}N{\times}L$) and its element |
| $\Delta M_{i',i,j}(t)$ | Incremental weight change at time $t$ |
| $\mathbf{E}(t)$ | Laplace states for EM timeline ($N{\times}N{\times}L{\times}L_e$) |
| $\sigma_l$ | Decay rate of EM integrator $l$ |
| $\tau'_l$ | Peak time of EM unit $l$; $\tau'_l = k/\sigma_l$ |
| $\tilde{\mathbf{M}}(t),\ \tilde{M}_{i',i,j,l}(t)$ | EM timeline tensor after inverse transform (four-tensor) |
| $\varepsilon$ | Index $(1 \ldots L_e)$ of episodic slice selected at retrieval |
| $\delta$ | Look-ahead interval used for future prediction |
| $\mathbf{R}^\delta$ | Diagonal time-translation matrix; $R^\delta_{j,j} = e^{-s_j\delta}$ |
| $\mathbf{p}(t+\delta)$ | Predicted input vector at $t+\delta$ from *standard* associative memory |
| $\mathbf{p}^\varepsilon(t+\delta)$ | Predicted vector at $t+\delta$ using episodic slice $\varepsilon$ |
| $\Delta t$ | Discrete simulation time-step (numerical integration) |

Table A1: List of notations used throughout the paper. Bold symbols denote vectors or tensors; subscripts $i, i', j, l$ select specific components.

