# OpenReview forum: "Computational Model for Episodic Timeline Based on a Spectrum of Synaptic Decay Rates"
_ccneuro.org/CCN/2025/Proceedings — CCN 2025 Proceedings asProceedingsTalkPoster_

### Official Review · Reviewer_3xto · 2025-03-28
**A biologically inspired model bridging working and episodic memory through multiscale synaptic dynamics**

**Soundness:** 2
**Clarity:** 2

**Comments:**

The paper is highly relevant to the CCN community. It addresses a fundamental question in computational cognitive neuroscience: how temporally structured episodic memories can be encoded, stored, and retrieved via biologically plausible mechanisms. It integrates insights from systems neuroscience, cognitive psychology, and mathematical modeling, and is thus interesting to broad audience.

By explicitly modeling both working memory and episodic memory as log-compressed timelines, and showing how synaptic changes with a spectrum of decay rates can store episodic events, the work bridges between short-term and long-term memory models. This contributes both conceptually and methodologically to a key research question in cognitive neuroscience.

That said, several important aspects remain underdeveloped. First, while the paper draws on biological observations (such as the spectrum of decay rates in synaptic plasticity), the link to actual neural mechanisms could be clarified. The biological plausibility of the system as a whole—especially the idea of maintaining and accessing a timeline of synaptic weight changes—remains somewhat abstract. The discussion on long-term potentiation is useful, but it stops short of addressing how such slow synaptic traces would be read out or stabilized in a functioning neural system. Moreover, the potential relationship to known oscillatory mechanisms in the hippocampus (such as theta-gamma coupling or phase precession) is left unexplored. Work like Lisman et al. (2005) on phase coding and sequence retrieval in CA3-CA1 circuits could provide valuable context, especially given the model’s emphasis on sequential structure and compressed time.

Second, although the authors build on the temporal context model (TCM) framework, the connection is not discussed in depth. How this model differs from or improves upon TCM in terms of psychological predictions and mechanistic assumptions would be important to articulate. For example, TCM also produces contiguity and recency effects—what specific experimental signatures could distinguish the current model from TCM or related frameworks?

The paper would also benefit from capacity analyses in terms of number of sequences that can be stored and their length, considering the well-known limitations of hebbian plasticity. Alternative approaches such as modern Hopfield networks or dense associative memory frameworks may offer more robust retrieval with controlled interference, and could serve as useful points of comparison.

On the behavioral side, the model appears capable of reproducing key psychophysical effects, such as the sublinear increase in retrieval latency with lag (recency), but the fit to data is largely qualitative. Formal simulations of response time distributions or comparisons with empirical results from recency judgment or free recall tasks would ground the model more firmly in experimental psychology.

The paper is clear. The progression from working memory to episodic memory is gradual and logically structured. The figures are effective, particularly those that show the compression of time, the learning of associations, and the distinction between episodic and working memory retrieval. The use of both orthogonal and non-orthogonal stimuli helps to illustrate the model’s robustness and limitations. Code is made available in an anonymous repository which facilitates reproducibility.

**Expertise:**

2

**Interest:**

3

---

> ### Author Rebuttal · Authors · 2025-04-15
>
> We thank the reviewer for excellent suggestions!
>
> **On comparison to earlier work and empirical results:** The proposed model builds on earlier work including the TCM and Laplace framework, both of which construct a single associative matrix (similar to Eq. 5 in our manuscript), where for TCM the memory is represented through a single leaky integrator while the Laplace framework (similar to the proposed model) uses a set of leaky integrators with a spectrum of time constants that are then inverted to represent information in the form of a timeline. The proposed model extends this approach to include a spectrum of synaptic time constants, which encode the history of synaptic changes, enabling temporal localization and retrieval of information from individual episodes. Empirical predictions generated by the proposed model indeed apply to tasks such as free and cued recall but when they are conducted over an extended period of time (e.g., when different lists are presented to participants across hours or days and when participants are provided semantic or temporal cues isolating a particular list). In the conventional tasks where the test follows the list presentation (immediately or after a brief delay), the proposed model could reproduce the existing behavioral results captured by the previous work (e.g., Howard et. al. 2015, Psych. Review) since it has an identical associative matrix. Please see lines 515-530 for an updated discussion on this.
>
> **On capacity analysis:** In the proposed model, for both working and episodic timelines, each input vector is encoded using temporal smoothing with unimodal basis functions whose width is proportional to the time since the input was presented. This is equivalent to log-compression in time (since on the log axis the basis functions would be equidistant and equally wide). This is a direct consequence of implementing the Laplace and inverse Laplace transform. Thus, the memory representation does not have capacity in a traditional sense of the number of input patterns that it can store, but rather introduces systematic temporal compression. We expanded this discussion and reflected on the comparison with attractor-based models in lines 496-514.
>
> **On more details about maintaining and accessing a timeline of synaptic weight changes:** We added a discussion on how the memory traces could be maintained and expanded the links to other work related to hippocampal memory encoding and retrieval (lines 439-451).

---

> > ### Comment · Reviewer_3xto · 2025-04-18
> >
> > _(This is TPC providing a pointer to the [**Official Comment**](https://openreview.net/forum?id=4naoku2Ak0&noteId=cEgJcvyDhR) from Reviewer 3xto starting with:_
> >
> > > Thank you for the thoughtful and...
> >
> > _With this **Rebuttal Comment** posted on behalf of the Reviewer, the Authors can respond with one final **Reply Rebuttal Comment** during the Author-Reviewer Discussion phase.)_

---

> > > ### Author Response · Authors · 2025-04-20
> > >
> > > Thank you again for your continued constructive feedback!
> > >
> > > To address the important suggestion about biological plausibility, we will expand the discussion on known biological mechanisms. Specifically, in the context of phase coding, we will point to earlier work on the Laplace framework (Shankar et al. 2016, Neural Computation) with more details that demonstrate how phase precession can be used to translate the neural activity of leaky integrators forward in time within a single theta cycle, creating a timeline of the future. That work is built on several specific assumptions supported by earlier studies. In particular, the size of place fields increases from the dorsal to the ventral end of CA1 (e.g., Jung et al. 1994, J. Neurosci.), the phase of theta oscillations advances from the dorsal to the ventral end like a traveling wave (e.g. Patel et al. 2012, Neuron) and the synaptic weights change as a function of the phase (Wyble et al. 2000, J.Neurophysiology). In the context of our manuscript, this means that theta cycles would correspond to sweeps through a set of values of variable \delta in Eq. 8. When making predictions based on a retrieved episodic state (Eq. 13), the theta-sweep mechanism could plausibly operate on the probing signal, which is the working memory state \tilde{f}​(t+\delta) from Eq. 8. This would mean that within a theta cycle, the retrieved past context is probed by a sweeping representation of the working memory states yielding a timeline of predictions specific to that retrieved episode.
> > >
> > > We will also reflect on theta-gamma coupling, which is certainly a relevant mechanism in the context of sequence learning, as Reviewer correctly pointed out. Theta-gamma coupling makes somewhat different predictions than the Laplace framework. This is because the Laplace framework cannot generate equidistant memory slots but only log-compressed memory representation. This makes it more similar to earlier work on timing models, such as spectral timing (e.g., Grossberg &  Schmajuk 1989) and consistent with the log-compression observed in behavioral (Weber-Fechner law) and neural data, such as log-compression observed in time cells (Cao et al. 2022, Elife).
> > >
> > > Regarding the compression and number of sequences that can be stored, the main challenge to the quantitative analysis is that our model does not have a capacity defined by a pattern count. Unlike attractor networks (e.g., Hopfield) that store a limited set of discrete patterns, our model represents a continuous history via log-compression. Instead, the fundamental limit is temporal resolution, which degrades logarithmically with time (\Delta T \propto T). This resolution, determined by the number of timeline units and the decay spectrum, determines how well events within older sequences or closely spaced sequences can be distinguished upon retrieval. The analysis of the impulse response properties (e.g., Eq. 4 and in the general section *Working memory timeline* and *Episodic memory timeline*) directly characterize the properties of the compression and impact of different parameters. Specifically, it shows how resolution (related to the impulse response width) degrades proportionally with time and how it depends on parameters like k.
> > >
> > > Thank you again - we very much appreciate all the helpful pointers and suggestions for future work.

---

### Official Review · Reviewer_CsHt · 2025-03-29
**The paper presents a novel method to store and retrieve episodic memory but lacks clarity in the exposition and motivation of the model.**

**Soundness:** 3
**Clarity:** 2

**Comments:**

This paper presents a mathematical model for combining working and episodic memory
systems, enabling both cue-based and temporal retrieval of episodes. The approach uses
Laplace transform principles to represent memory in a time-dependent fashion, allowing
the disambiguation of episodes with identical cues based on temporal information during
a recall process in an intriguing way.

### Interest

The paper's novel theoretical contribution to memory modeling is
storing memory encodings temporally, as well as associations between memory encodings,
and retrieving related episodes by time shifting those representations. However, the
highly technical presentation limits accessibility for a broader Computational
Cognitive Neuroscience audience in the current form. Is there a concrete example of episodic memories that
could be used both in the introduction and the model exposition? This could tie the sections
together and demonstrate wider appeal.

Part of a broader appeal would be the alluded to neural implementation.
The papers discussion on this shows potential but remains incomplete. While
evidence for different synaptic timescales is quickly reviewed, the paper doesn't explain how
multi-timescale synapses would implement the described representations (f tilde, M tilde).
How would the time-translated versions of the memories be calculated in a network model?
How would a network model switch between "storage" mode and retrieval / prediction mode?

### Clarity

The mathematical exposition presents equations in detail but fails to communicate their
significance and computational implementation implications. I found several key
conceptual points confusing, particularly how "retrieval" equates to "prediction in the
Laplace domain." Please clarify in the text.

The associative memory tensor is composed of standard associative memory matricies
for each of the input dimensions, correct? Furthermore, why does f tilde have a second
dimension according to the text (line 272)? Please clarify in the text.

The visual presentation contains numerous issues. The paper has a lot figures,
making it difficult to compare related results. The representation of vector components
using three colored bars confuses rather than clarifies understanding. A clarification
in text may be enough, but I think a different visualisation would be more effective.

I found it  particularly challenging to identify the retrieval/Laplace prediction results in the
plots, especially in non-orthogonal cases and when memory was mapped back in time.
Is there a way to disambiguate the overlapping lines?

Figure captions provide insufficient explanation of what each experiment demonstrates
and how it supports the authors' claims. Combined with the high number of figures,
I would suggest merging some of the figures so that they support clear statements about
the model. This could be supported with a corresponding clear figure description.

### Soundness

The technical insight storing past inputs as filter responses interpretable as Laplace
transforms, and then adding associative memory on top of that is very interesting and theoretically sound.
The matrix-vector multiplication approach to memory retrieval
in the Laplace domain also works. But the claimed results are
difficult to verify from the presented figures, particularly in non-orthogonal cases
and when memory was mapped back in time. Perhaps a log-scale on the x-axis for the prediction
or something like this could clear this up? Further, is there a good intuition how reliably
the predicted / retrieved memory traces would be translated back to the original representation?
Even with the simple data used in the paper, I have difficulty understanding the curves and estimating
how good the method would work in practice.

### Suggestions

1. Accessibility: Provide less abstract examples demonstrating practical model
applications to make the work more accessible to a broader CCN audience.
2. Implementation Details: Clarify the neural implementation pathway, even if using
an artificial network model rather than a fully biologically plausible one.
3. Figure Organization: Combine related figures to facilitate direct comparison
between experimental conditions.
4. Conceptual Clarity: Clearly explain the equivalence between "retrieval" and "
prediction in the Laplace domain" earlier in the paper.
5. Visual Representations: Consider using semantic labels for stored vectors
instead of the current abstract color coding system.
6. Figure Captions: Provide more detailed captions that clearly explain what each
experiment demonstrates and how it supports specific claims.

**Expertise:**

2

**Interest:**

2

---

> ### Author Rebuttal · Authors · 2025-04-15
>
> We thank the reviewer for excellent suggestions!
>
> **On accessibility/concrete examples:** We added simulations with realistic sequences of images, which demonstrate the capability of the model to separate individual episodes and use of both temporal and semantic cues to retrieve information about a particular episode (Fig. 8 and A3). The sequences consisted of partially overlapping episodes (e.g., waking up in a hotel room in different cities). The model combines semantic (e.g., being in a hotel room) and temporal information (e.g., a year ago) to construct a temporally localized prediction (prediction based on the associative memories formed around the cued time). The output of the prediction is then a semantic vector with high similarity to the correct city.
>
> **On implementation details:** We added a complete pseudocode to the Appendix (in addition to the actual code updated with new examples) that describes how exactly the spectrum of synaptic timescales implements the episodic timeline and what is necessary for encoding and retrieval. In brief, the encoding process requires synaptic weights to act as leaky integrators (integration of inputs while gradually decaying), while the retrieval process requires matrix multiplications, commonly referred to as probing. We added a discussion on this in lines 439-451.
>
> **On visualization:** To address Reviewer’s concerns about visualization and captions, we changed the visualization layout in new figures (Fig. 8 and A3) and provided more detailed captions. Following a suggestion from the Reviewer, we grouped experimental conditions in single figure (see e.g., Fig. 8).
>
> **On conceptual clarity:**  To make the methods section more accessible, we rewrote the equations in a different format and separated the continuous (analytical solution) and discrete (implementation) forms. In general, to help with the intuition, the associative memory tensor stores associations between the input vector (length L) and the two-tensor (f tilde), which two dimensions encode what happened when (size L x N ). So, for instance, given a sequence *A, B, C*, at the time when *C* is at the input, the associative memory will associate *C* with *B* one time step in the past and with *A* two time steps in the past. Thus, the next time *A* is presented, that same association indicates prediction for *C* two steps in the future (this is a bit of a simplified illustration since, in our model, the time is log-compressed).

---

> > ### Comment · Reviewer_CsHt · 2025-04-20
> >
> > Thank you for the the significant revisions and additional information in the paper, my comments were largely addressed.
> > Due to improvements in the text and the added figures and experiments, I'd like to specifically upgrade my previous ratings to "Disciplinary" in the interest category and to "Strong" in the Soundness category.
> >
> > I think the new Figure 8 has some weaknesses in the combination of description and readability (for example including figure part labels for reference), but I think its inclusion still improves the text, the experiment is very impressive.
> >
> > Overall, I'd like to thank you again for a very interesting perspective on episodic memory and I'm looking forward to models inspired by this idea.

---

> > > ### Author Response · Authors · 2025-04-20
> > >
> > > Thank you very much for your feedback and encouragement! We will make sure to improve the readability of Figure 8 in the final version and include figure parts labels for different rows - we agree that this will improve the clarity.

---

### Official Review · Reviewer_TCdC · 2025-03-31
**A Theoretically Intriguing “Log-Transform” Episodic Memory Model, could benefit from More Empirical Grounding and Rigorous Parameter/Noise Analyses**

**Soundness:** 3
**Clarity:** 2

**Comments:**

I would like to put a disclaimer upfront. I might not possess the expertise required to evaluate the submission and I sought assistance from several mentees from my group (with CS, or data science, or maths background) and a computational colleague for this review. In the following, I compiled our reviews for your reference. Overall, I rate the submission highly for the three aspects (interest, soundness, and clarity) and provided some detailed comments in the following.
The authors propose a two-tiered memory model: (1) a working memory timeline using exponentially decaying neural activity (log-spaced time constants) to approximate the Laplace transform of recent inputs, and (2) a slower set of decays for episodic memory, capturing changes in synaptic weights over longer periods. By “inverting” these transforms, the system can reconstruct past states and retrieve distinct episodes, thus avoiding the superposition problem of standard associative memory.
Comment 1
Strengths
1. Mathematically Elegant Concept Employs well-known Laplace transform theory (and approximate inversions) to encode a continuous, scale-invariant temporal representation. (Side Note: This aligns with prior work on “Laplace codes” for time, giving strong theoretical underpinnings.)
2. Unified Timescales Demonstrates a single conceptual framework for bridging short-term (seconds/minutes) and longer-term (hours/days) memory storage, simply by using multiple (log-spaced) decay constants.
3. Improvement Over Basic Associative Models Shows that by storing the history of weight changes rather than final weights, one can retrieve an individual episode—rather than a smeared average. The figures showing how the system “jumps back in time” to retrieve specific sequences are compelling.
4. Biological Motivation References multiple phases of synaptic plasticity (E-LTP, L-LTP) to justify the “spectrum” of decay rates. *(Side Note: the notion that synapses may naturally span from short hours to long days is consistent with real data.)
Critiques
1. Over-Reliance on Simplistic, One-Hot Inputs
All demonstrations use toy examples with orthogonal stimuli. Real data often involve correlated features or embedded vectors with partial overlaps. (For ML/CS, we frequently test memory models on more realistic vectors—e.g., even simple textual embeddings. One-hot vectors might inflate success rates by avoiding feature collisions.)
2. Minimal Empirical/Behavioral Validation While the paper alludes to known phenomena (temporal contiguity, sublinear scaling), it never actually simulates free-recall tasks or compares to human data. “Jumping back in time” is shown in a purely toy environment. No direct link to known recall patterns (e.g. lag-recency curves).
3. No Parameter Sensitivity or Noise Analysis The authors assume an ideal log-spacing of time constants but do not explore whether slight deviations or fewer/more constants break the systemʼs retrieval. Synaptic noise or drift—expected in biological systems—is omitted. (This is significant. If the system is extremely sensitive, thatʼs a major drawback for real neural implementation.)
4. Sparse Details on Biological Implementation The text claims “biological plausibility,” but the actual circuit-level explanation of how to compute the inverse Laplace transform is missing. The authors do not discuss iterative or approximate methods that real neurons might use (and do not handle partial errors in reconstructing older events). (The “spectrum of LTP phases” is interesting, but we also need a local plasticity rule to calibrate the inverse transform. This is not described.)
5. Multiscale Terminology Might Be Overstated “Multiscale” usually implies a more intricate or hierarchical approach. Here, itʼs just repeated exponential decays on a log axis. While valid mathematically, calling it “multiscale synaptic decay” could mislead some readers looking for multi-layer or manifold-based structures.
6. Lack of Large-Scale or Continuous Test Cases Examples mostly revolve around storing a small handful of discrete sequences. Potential issues like interference, “blending” with many episodes, or capacity limits are not addressed. Real neural systems often handle extensive streams of ongoing experience.
7. Equation Clarity & Derivation The authors reference Postʼs inversion formula but do not show how itʼs discretized or how many derivative approximations are used. A more hands-on explanation (maybe a pseudo-code snippet) for the feedforward inverse layer would improve reproducibility.
Suggestions for Improvement
1. Test More Complex/Realistic Input Space Provide at least one example with correlated or continuous features—this is crucial to show the modelʼs viability beyond toy one-hots.
2. Include Noise & Parameter Studies Show how small changes in time-constant spacing or additive noise in synaptic updates affect retrieval. This clarifies the range of conditions under which the model remains functional.
3. Compared to Empirical Memory Data Even a basic free-recall simulation (does the model reproduce recency/contiguity curves?) would substantially raise confidence that these ideas map to reality.
4. Expand on Biological Implementation Offer a plausible scenario for how circuits might approximate the derivative based inverse. Maybe partial replay or iterative top-down feedback? Acknowledge potential learning rules for calibrating these “decay constants” and feedforward weights.
5. Clarify “Multiscale” in the Title Emphasize that it means multiple exponentials along a log axis, not necessarily a hierarchical or multi-layer approach. Avoid confusion with other “multiscale” usage in neuroscience.
6. Explicit Pseudo-code or Algorithm Summarize the forward (encoding) and inverse (decoding) process step by step. This helps outside readers replicate or build on your method.
The paper presents a conceptually elegant system for bridging short-term and episodic memory via log-spaced decays and Laplace transform inversions. The potential for addressing “which episode happened when” is appealing—especially for artificial or computational frameworks that need a compressed timescale code. However, the lack of realism in input domains, absence of thorough parameter/noise analysis, and scarce empirical grounding limit the paperʼs current impact. Strengthening those areas would make this work far more compelling for the CCN audience and better substantiate the ambitious claim of a multiscale synaptic memory mechanism.
Comment 2.
The manuscript provides an approach of building the computational model of the encoding and decoding of working memory and episodic memory based on Laplace transform and Reinforcement learning. The model itself obeyed a rigorous deduce process, and a fine logic flow is constructed. The result is solid and align with the theory.
Some inquiries:
1. The reason of using RNN in the process
What are the pros of Reinforcement learning in executing Laplace transform against other methods?
2. The relationship between model for working memory and episodic memory
In the article, the authors seem to build the working memory model as a lower level model than the episodic memory. However, it is questionable that characteristic of working memory are fully represented in the working memory model, and some difference between working memory and episodic memory may be ignored.
3. Episodic memory model
For the illustration of the “jump back in time” in EM model, the $\tilde M$ in the figure is a little bit confusing without explanation in captions.
Comment 3.
The paper contains a link to a repository with code, which is a good thing as it allows others to reproduce the figures. However, the mathematical formulation of the model in the text is very sparse, and its current form would not allow the reader to code up the model by themselves.\\
It is not clear what method is used to {\emph{analyticaly compute}} the recurrent weights in l. 235. The paragraphs around Eq. (1) do not allow the reader to deduce the equation. More explanation would be needed to motivate the equation: how are the general properties of recurrent networks and feedforward layer used to write it down? Perhaps a toy model with a very small network could be used to demonstrate the construction of the model.\\
 The Laplace transform would need a gentler and more explicit introduction, with a formula defining it in terms of the function of time, and the variable in the Laplace domain. The variable in the Laplace transforms is often denoted by $s$, and it has the unit of a frequency (or rate), but how is this rate related to the components of the vector $\mathbf{s}$, described as "{\emph{a vector composed of $L$ log-spaced values that determine the rate of decay of units}}. In Eq. (1), this variable seems to be a vector denoted by $\mathbf{s}$, whereas in Eq. (3) the quantity $s$ entering the quantity $\tau^\ast$ is a scalar. How are these two quantities related? Besides, the object $\tilde{f}^i_t$ would deserve another name if it corresponds to a special choice of $f_t^i=\delta(0)$.\\
 In Post's inversion formula for Laplace transforms, the r.h.s. is the large-$k$ limit of Eq. (3) (see {\ttfamily{https://www.rose-hulman.edu/~bryan/invlap.pdf}} for a proof). Technically, a limit symbol is missing from Eq. (3). Moreover, putting the formula to use requires choosing a sufficiently large value of the integer $k$. This is an additional parameter of the model, which needs to be discussed.
Apparently, the code uses the value $k=16$ (a comment in the code indicates that {\emph{larger $k$ causes more narrow peaks in $\tilde{f}$}}). This comment should be expanded and be made part of the main text: it seems to explain that larger values of $k$ would be too large, but the reader needs an argument to justify why $k=16$ should be considered high enough in the first place (indeed, from Post's formula one would expect larger values of $k$ to yield better results).\\

**Expertise:**

2

**Interest:**

3

---

> ### Author Rebuttal · Authors · 2025-04-15
>
> We thank the reviewer and colleagues for excellent suggestions!
>
> **On testing more complex inputs:** We added simulations with realistic sequences of images. We computed CLIP embeddings and fed them into the proposed memory model. The model was able to separate individual episodes and use both temporal and semantic cues to retrieve a particular episode, unlike previously used associative memory (Fig. 8 & A3).
>
> **On including noise:** This important consideration has been extensively studied in previous work. We added a section in the Appendix about how the results from Shankar & Howard 2012, Neuro. Comp. (Chapter 4 Effect of Noise) translate to the current work.
>
> **On comparing to empirical data:** The proposed model expands earlier work which uses a single associative matrix, similar to Eq. 4. Earlier work has demonstrated the ability to account for results in free recall (e.g., Howard et. al. 2015, Psych. Review, Fig. 8). The proposed work introduces a timeline for episodic memory, inheriting the properties of the Howard et. al. 2015 model and expanding them by separating distinct episodic memories. This creates empirical predictions for tasks such as free recall when they are conducted over an extended period of time (spanning hours or days) (lines 515-530).
>
> **On biological implementation:** We provided additional details in lines 452-469. We referred to recent work on how the weights needed for implementation of the Laplace transform could be learned (Alipour et al. 2025, J. Comp. Neuro.) as well as earlier studies on the plausibility of the circuit-level models for the inverse Laplace transform (e.g., Liu et al. Hippocampus, 2019).
>
> **Responses to other suggestions:**
>
> We changed “Multiscale” to “Spectrum” in the title.
>
> We added the pseudocode to the Appendix and changed the equation style to make it more accessible.
>
> We used an RNN to ensure time-locality of the model (e.g., transformers would not be time-local).
>
> We did not use a reinforcement signal but only associative memory (Eq. 5).
>
> We expanded the discussion on the relationship between the models for working and episodic memory (Intro and Discussion), added figures with detailed captions (Fig. 8 and A3) and expanded existing captions (e.g, Fig. 5).
>
> We changed the format of the equations and tried to make the intro to the Laplace transform more accessible. We described the properties of variable **s** (lines 249-250).
>
> We expanded the discussion on parameter *k* (lines 278-283).

---

### Meta-Review · Area_Chair_n9F4 · 2025-05-05

**Ccn Recommendation:** Accept as Proceedings

**Metareview:**

The reviewers deemed the work of high relevance for the CCN community although some concerns were raised about the accessibility of the text for the general CCN audience.

Novelty: While the idea of using the Laplace transform for multitimescale codes has been successfully used in the neural domain and as part of multiscale reinforcement learning, the focus on synaptic decays is novel. There are potentially unexplored with palimpsest models of complex synapses (e.g. Benna Fusi 2016) even if the mathematical parallels are not very tight.
Approach: the methodology seems sound and the claims are supported by the results.
Significance: it is not clear what are explicit neural and behavioral signatures of this kind of code that could disambiguate the model from alternative episodic memory systems (e.g. the scaffolding work from Ila Fiete's group).
Clarity: the text remains somewhat dense in places but understandable for within domain part of the audience.
Overall, I feel that there is enough to commend the model to support its presentation in this current form.

**Summary:**

The paper proposes a novel computational model for storing temporally organized episodic memories, in the form of a neural representation of "what happened when". This is mechanistically mediated by a basis of different synaptic decay time constants, formally translating ideas about Laplace transform-based multiscale temporal representations to the synaptic domain.

**Expertise:**

3